

2                                                                                     Ocean Science

3                                                                                     June 1, 2020.

**6        Technical Note: Estimation of global loss of freshwater based on sea level**

**7                                 changes over geological time**

9                                                Gaspar Banfalvi

Department of Molecular Biotechnology and Microbiology, University of Debrecen,

1 Egyetem Square, Debrecen 4010, Hungary

**Correspondence**:

Prof. Gaspar Banfalvi

University of Debrecen

Department of Molecular Biotechnology and Microbiology

Life Sciences Building 1.102

1 Egyetem Square, Debrecen 4010, Hungary

Tel. (36) 52 512 900 ext. 62319;

Fax: (36) 52 512 925

Email: gaspar.banfalvi@gmail.com

bgaspar@unideb.hu





**Abstract**
Water vapour at the upper layer of the atmosphere undergoes light-dependent photolysis generating
reactive hydrogen ions *in statu nascendi*t hat escape to space by different mechanisms. Besides
hydrogen, other volatile gases such as methane molecules and helium atoms also escape to space in
smaller quantities or traces such as oxygen. The escape of hydrogen through the planetary air leak
cannot be reliably judged. Our estimation of global freshwater loss used another approach based on the
sea level changes that continuously fluctuated over geological time. The most reliable evidence for
eustatic sea level changes was provided by geologists estimating the shifts of shorelines generating
sedimentary deposits. The sea level changes turned to volumetric data of a) radii of the Earth ($r_1$) to
calculate the volume of the geoid Earth ($V_1$) comparing and validating them with available
estimations, b) average sea depth ($r_2$) comparing volumetric values of the sea with best-fitting values
($V_2$). c) showing the correlation between geological changes ($r_3$) and corresponding sea volumes ($V_3$).
These data,along with the sea volume of the infant Earth, allowed to plot a calibration curve to
calculate the sea level belonging to the volume and *vice versa*. Geologic data indicate the
shrinkage of freshwater pools during interglacial dilution periods and the remarkable long-term
salination of the ocean.

**Keywords**
water vapour, photolysis of water, H escape, freshwater reserves, sea level rises, salination of
sea










## 1. Introduction


Water vapour at the upper layer of the atmosphere undergoes light-dependent photoionisation
generating reactive hydrogen ions *in statu nascendi* that escape to space by different mechanisms. The
escape velocity of hydrogen to the outer space is the highest among light gases followed by helium,
ammonia and water vapour and is dependent on gravity, which in the inner planets of our solar system
is lowest on Venus followed by Earth and Mars.The escape of gases, among others, is dependent on
the temperature, rate of cooling, greenhouse effect. It was estimated that the Sun would be brighter and
hotter in about 1 billion years and the global surface temperature could rise to about 47 $^{o}$C. The Earth
will be likely to lose its water (Schröder et al., 2008) similarly to what already happened to Venus.
Losses of light gases are unlikely to be inherent parts of the evolution, at least in the inner planets of
our solar system. Vast losses of gases also came from the activity of the young Sun that blew light
gases, primarily hydrogen and helium to the outer planets known as gas giants. An important factor
related to the global loss of water is the polar wind that drives hydrogen and other ions to space seen
as faint yellow area near the Earth's poles.
The shrinkage of freshwater reservoirs including atmospheric moisture (snow, rain, clouds) and ice
(glaciers, polar ice, ice sheets, permanent snow) is contributed by the freshwater loss of global
warming. The atmospheric escape of hydrogen on Earth is assumed to result in approximately 3kg/s
loss of hydrogen and about 50 g/s loss of helium (Zahnle, 2006; Catling and Zahnle, 2009).The
abundant $CH_4$ originating from methanogenesis could have supported the escape of hydrogen to space
by orders of magnitude faster than today (Catling et al., 2001).Our own repeated calculations showed
that under the conditions that exist today, the H escape could have resulted in only about 0.02% loss of
the recent ocean volume. The explanation to this negligible loss could be that the escape of water: a)
was much faster earlier, b) came from different sources that could sum up or c) was not significant
during the evolution of Earth.
Regarding oxygen, only small quantities were found to escape to the Moon from the Earth (Terada
et al., 2017). $O_2$ produced by photosynthesis absorbed in oceans and seabed rocks started to gas out
about 1,850 Mya but was absorbed mainly by land surfaces. When the ocean saturated, oxygen began
to accumulate in the atmosphere (Holland, 2006).





An important warning signal of freshwater loss is the increasing gap between saltwater and
freshwater, which is now 97 *versus* 3 %. The question is how this ratio could have changed from a
dilute too salty and concentrate sea during evolution. The highest known sea level some 3,700-3,800
million years ago, suggested that the Infant Sea could have contained 26% more water than today
(Pope et al., 2012; Rosing et al., 2010). Ice ages generated low sea levels particularly during the
Proterozoic Snowball Earth period when Earth's surface became almost entirely frozen and covered by
slush, snow and ice (Kirschwink, 1992; Allen and Etienne, 2008; Pu et al.,2016). The opposite
tendency, namely the accumulation of substantial freshwater reservoirs was assumed to take place in
the late Proterozoic aeon before the Cambrian period (541 Mya) and between the Ordovician and Silur
epochs (Pu et al., 2016; Haq et al., 1988; Holland et al., 1986). Severe glaciations with low sea levels
were followed by extreme high sea levels, *e.g.* during the Paleozoic era at the end of the Ordovician
period (~450 Mya). Although the Ordovician lasted for only 45 million years and represented only
~1% of the age of the Earth, the life on Earth during this time diversified rapidly. The unprecedented
radiation of species in Ordovician is accounted for by the dilute, yet optimal osmolarity (0.2 – 0.4
Osm) of the oxygenated ocean and the abundance of freshwater supply.
Gas escape theories did not provide reliable means to judge the global loss of water on Earth.
Thus geological sea levels were turned into volumetric changes to give a reasonable explanation to
the shrinkage of freshwater.
**2. Assessing freshwater loss**
**2.1. Estimation of sea volume and turning sea levels to volumetric data**
We used calculations of interconversions to turn sea levels and volumes into each other.The validity
of the data obtained was tested by using:

*i)*    Geometric radii of the geoid Earth and average radius

*ii)*   Reliable estimates of average sea depth

*iii)*  Comparison of volumetric estimates and selection among best-fitting values

*iv)*   Testing whether volumetric changes belonging to sea levels can be used to construct a

calibration curve





*v)*  Plotting the calibration curve to show the relationship between sea level rises and volumetric

changes in sea volumes (Banfalvi, 2017).

**2.2 Sea volumes**

The extention of the calibration curve included the highest known sea level of the sea of the Infant
Earth (~ 750 m) relative to the recent sea level (6,371,008 m). Compared to the average radius of the
Globe (6367.3 km), the average depth of the sea (3682.2 m) is negligible thus the linearity of the
calibration curve is not questionable (Fig. 1). Most of the data of the sea volumes were 30-40 years
old, differed from each other and were significantly higher than the recent satellite measurements
(https://www.livescience.com/6470-ocean-depth-volume-revealed.html). One could not explain the
higher estimates of current sea volumes by recent changes. Rather, the amounts of undersea
mountains, ocean ridges and other geographical features were not subtracted from the bulk of the
ocean resulting in higher estimates. These structures under the sea level include *i)* the Globe itself, *ii)*
continental shelves, slopes, rises, insular shelves that surround continents and islands, and *iii)* zones of
the ocean floor (continental margins, deep-ocean basins, mid-ocean ridges, sediments) that are merged
and referred to as undersea features.

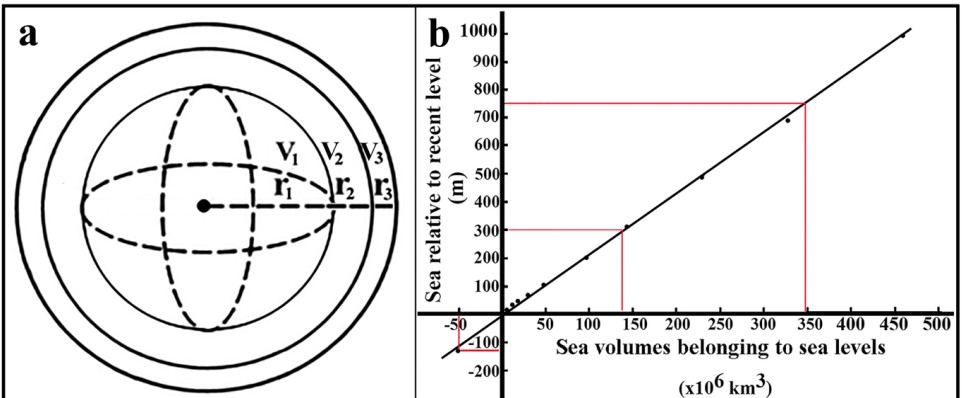

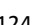

Figure 1. Demonstration of sea levels and volumes (a), plotting them into a calibration curve (b).

a) Data used for the calculation of underwater volumes of the continental shelf and other underwater

features include: $r_1$ = average of equatorial and polar radii of geoid Earth (6367.3 km) (Banfalvi,


2017). $r_2$ = radius of Earth + depth of sea (6,371,008 m) (Williams, 2012). $V_1$ = volume of Earth
calculated from $r_1$ (1,081. 320 x$10^9$ km$^3$). $V_2$ = volume of sea (1332.9 x $10^6$ km$^3$) (Banfalvi, 2017).
$V_3$ = actual volume arising from sea level rise (0.09 m x 361.84 km$^3$ = 32.5 km$^3$). b) Relationship
between sea level elevations and sea volumes relative to recent data. The data of abscissa represent
eustatic sea level elevations and those of the ordinate the volumetric changes relative to recent data.
Lowest sea level (-130 m) (~20,000 years ago), higher (~300 m) (~500 million years ago) and the
highest sea level of the Infant Earth (3,700-3,800 million years ago). Modified with permission
(Banfalvi, 2017).

Table 1 shows the recent volumetric data of water on Earth (Table 1). The continental margin is

constituting about 28% of the oceanic area (Cook and Carleton, 2000). All the underwater features
represent an estimated 29.5 % of sea volume and will be more precisely determined as the
measurements will become eeven more accurate. The most reliable data obtained by satellite
measurements of sea surface face some light reflection problems (Charette and Smith, 2010). The
resolution of the sea depth needs further "fine-tuning" by ship-based echo sonar and other methods.
Satellite measurements show that the average ocean depth is 3,682.2 m. This value multiplied by the
area of sea (361.84 x$10^6$ km$^3$) results in the volume of the ocean of 1,332.4 x$10^6$ km$^3$ (Charette and
Smith, 2010). These measurements show a good match (±0.1%) with other values published within
the last ten years (Table 1).
Table 1
Volumetric data of water on Earth

| # | Volume of | Volume x $10^6$km$^3$ | Volume (%) | References |
|---|---|---|---|---|
| 1 | *Sea | 1332.4 | 96.9 | Charette and Smith, 2010 |
|   |   | 1332.9 | 96.9 | Banfalvi, 2017 |
|   |   | 1335 | 97.1 | Eakins and Sharman, 2010 |
|   |   | 1335 | 97.1 | Durack, 2015 |
| 2 | **Sea (average) | 1333.8 | 97.0 |   |
| 3 | [†]Freshwater in reservoirs | 25.54 | 1.86 | Durack, 2015 |
| 4 | ‡Freshwater in land | 15.66 | 1.14 | Durack, 2015 |
| 5 | Freshwater total (#3 + #4) | 41.20 | 3.00 | Durack, 2015 |
| 6 | Water on Earth (#2+#5) | 1376.2 | 100 | Durack, 2015 |
| 7 | Sea (Infant Earth rel. to #2) | 1681.1 | 126 | Banfalvi, 2017 |
| 8 | Global loss of water (#7-#6) | 305 | -22 | This estimate |


*Sea volume, average of four estimates rounded up to 1335 x $10^6$ km$^3$ (70.015%).
†Freshwater reservoirs: ice, glaciers, permanent snow.
‡Freshwater inland: lakes, artificial lakes, ponds, streams, wetland and groundwater.
Estimation of the global loss of water was obtained by subtracting the recent volume of the sea from
the volume belonging to the highest sea level.
The largest volume represented by the volume of Earth + volume of the sea. Calculated from $r_2$:
6,371,008 m → 1,083.210 x$10^9$ km$^3$.
Volume of underwater features including continental shelf. Substraction of volumes calculated from $r_2$
and $r_1$: 1,083.210 x $10^9$ km$^3$ - 1,081.320 x$10^9$ km$^3$ → 557 x $10^6$ km$^3$ (29.5%)
Summary of volumes of sea 1332.9 x $10^6$ km$^3$ (70.5%) plus underwater features and continental shelf:
557 x $10^6$ km$^3$ (29.5%), total 1889.9 x $10^9$ km$^3$ (100 %).
Calculation of continental shelf: land surface area of the World  x average depth of ocean: 148.42 x$10^6$
km$^2$ x 3,6822 km = 548.42 x $10^6$ km$^3$.
Volume of underwater features without continental shelf: 557 – 548.42 = 8.57 x $10^6$ km$^3$ = 0.45%.

Our calculations showed that the volume of the underwater features excluding the continental shelf

represents only 0.45 % of the size of the ocean, confirming that the average seabed (ocean floor) is a
relatively flat surface. John Murray (1888) utilised a simple model and measured the depth of the
ocean at several locations then calculated the ocean volume by only 1.2% higher than the current
estimates of the amount of the world's oceans (Charette and Smith, 2010). Our data showed a close
relationship between sea level rises and volumetric changes in the sea. These data were useful to
create a calibration curve (Figure 1) (Banfalvi, 2017). The calibration curve was then extended to
apply it to the highest sea levels. More importantly, the calibration curve served to estimate global
water loss.

The elevations of sea level rises originate mainly from the melting of freshwater reservoirs and

thermal expansion during interglacial periods. Predictions forecasted that the recent interglacial
period could melt about 80% of the Earth'sice and snow reserves of ~50 x $10^6$km$^3$ (Berger and
Loutre, 2002). Others judged that the freshwater reserve is only about half (24 x $10^6$ km$^3$) of the
earlier estimation (Shiklomanov,1995).The newest estimate of freshwater is somewhat higher (25.54
x $10^6$ km$^3$) (Durack, 2015) but only about half of the maximum of the latest ice age some 120,000
years ago. By fitting sea level rises and volumes into the calibration curve (Fig. 2), sea levels can be
turned into volumetric values and *vice versa.*



The highest reported sea level was at the period of the Infant Earth (Pope et al., 2012; Rosing et
al., 2010). To estimate the global loss of freshwater, the highest sea level rise of the Infant Earth
served as a basis. This sea level could have been by 750 m higher than it is today (Fig. 1b) with a sea
volume of ~ 1,681 x $10^6$ km$^3$. Volumetric data of sea and freshwater are summarised in Table 1. The
calibration curve (Figure 1b) takes into consideration that in the presence of continental crust, the sea
level is about 29.5 % higher than in its absence. Due to the lack of landmasses, the early Earth was
assumed to be completely covered with water (Rosing et al., 2010). Thus the sea level of the Infant
Earth could have been roughly 530 m higher than it is today.
**3. Geochemical stability *versus* dynamic nature of global osmolyte system**
Despite large sea level fluctuations, only small changes can be traced if at all, leading to the
conclusion that in general, the rate of input and output in the sea was nearly equal in agreement with
the long-term mean of its salt concentration (Pope et al., 2012). The idea of a general geochemical
balance of the sea is related to the limited foreseeable future of man that is not longer than 100 years
but provided a model to make such constancy plausible (Rubey, 1951; Railsbeck et al., 1989). The
view of a dynamic osmolyte rather than a steady-state ocean system is gaining ground, by measuring
short-term volumetric decreases during ice ages; sea level rises during interglacial periods and long-
term salination of the ocean (Banfalvi, 1991; 2016;  2017).
The    Unified    Sea    level    Rise    Projection    (http://southeastfloridaclimatecompact.org/wp-
content/uploads/2015/10/2015-Compact-Unified-Sea level-Rise-Projection.pdf) serves as a reasonable
source of information to predict the sea level rise in the 21st-century relative to the 1992 mean sea
level. The short-term rise envisaged 15-25 m by 2030. The medium-term sea level rise projected 25-66
m with a less likely possibility of reaching 86 m by 2060.The long-term projection predicted nearly
79-155 m sea level rise and a less likely extension to 206 m (Southeast Florida Regional Climate
Change Compact, 2015). The melting of 20 x $10^6$ km$^3$ ice and snow would correspond to about 80%
loss of  the available freshwater reservoir and would reduce the global freshwater volume to ~1.51%
from 3% and cause a sea level rise of about 50 m (Figure 1). A higher sea level rise is less likely to
occur than that predicted by 2060 in the Unified Sea level Rise Projection. Nevertheless, it may
correspond to the forecast of the Intergovernmental Panel on Climate Change (IPCC) 5th Assessment



Report (ARS5) model projection (Nerem et al., 2018). It is unrealistic to assume the complete melting
and loss of the freshwater reservoir during the recent interglacial period. A higher than 50 m sea level
rise is unlikely to be reached simply because of the shortage of the freshwater reserve. The complete
melting of the freshwater reservoir would be catastrophic, especially to the terrestrial vertebrates that
are fully dependent on freshwater supply. Forced sea level rise patterns were predicted and could
continue in the coming decades with elevated rates of rises (Fasullo and Nerem, 2018), but the rise
may not be linear with time. The initial faster rate of sea level rise could slow down due to the
shortage of freshwater reserve.
**4. Conclusions**
One explanation for the global freshwater loss is that water vapour at the upper atmosphere
photohydrolysed to H and oxygen. H escaped to the outer space; oxygen formed the protective ozone
layer. Calculations showed that the recent escape of hydrogen to space is low, but it could have been
much higher at the earlier periods of global history. Calculations related to the hydrogen escape theory
did not allow far-reaching conclusions regarding a significant water loss. Thus evolutionary changes
of sea levels and their converted data to sea volumes were used to calculate the loss of water on Earth.

The consequence of gradual loss of water on Earth is that the salt concentration of sea increases.

Based on Raoult's law applied to the sea as a global osmolyte system, the evaporation of water vapour
decreases and results in less precipitation and freshwater.

Characteristic biological phenomena accompany the loss of freshwater. Although the recent sea

level rise will not significantly reduce the salinity of the sea, due to the limited freshwater reserves, its
effect will severely impact flat seashores impacting many large cities. The life of sea animals will be
hardly affected, unlike land vertebrates, including man that are entirely dependent on freshwater. The
migration of people driven by the shrinkage of available freshwater and the spread of deserts are
continuing. Due to the freshwater shortage, the habitat of species is decreasing at an alarming rate,
threatening with extinction many endangered freshwater species.

It is concluded that different sea levels with decreasing heights during evolution can be used to

estimate the global loss of water. The sea level of the Infant Earth could have been by 26% more
voluminous than it is today. The second-largest sea levelrise (~300 m) took place some 500 million



years ago. Fluctuations indicate record-low sea level (-130 m) about 20,000 years ago. The basis of
estimation of the global loss of freshwater is the subtraction of the recent volume of the sea from the
amount belonging to the highest sea level. The substantial freshwater shrinkage contributed by man
and the salination of sea demand counteractions to be taken or already in effect to protect life on Earth.
**Data availability**
The MS will be deposited into the institutional repository of DEA (Educational and Research
Support) of the University of Debrecen and National Library.
**Author contribution**
All activities related to the preparation and publication of the manuscript were carried out by
the single author.
**Acknowledgement**
This work was supported by the OTKA grant T0 42762 to G.B.
**Conflict Of Interest Statement** The author declares to have no conflict of interest.
**ORCID**
*Gaspar Banfalvi* https://orcid.org/0000-0002-6304-7653
**Ethical consideration**The author has no ethical issues to be considered.

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
