# Peer review of "(untitled)"

_Ocean Science, 2020_

## Referee Comment (RC1) · Anonymous Referee #1 · 23 Jun 2020

While I assume that the author is sincere in his beliefs, it is difficult to make any sense of this manuscript. Unfortunately I cannot even summarize the main purpose of this paper, although it appears that somehow changes of sea level of up to 750 meters are to be related to...I am not sure. Eventually a reference is made to the ocean being a 'dynamic osmolyte', whatever that is, with reference to what appear to be earlier papers with much the same calculations in something that appears to be an on-line scam journal, and a claim that global sea levels could rise by 20 meters by 2030, with reference to a data report published by a south-east florida working group that reproduced some IPCC report figures that showed a possible rise of perhaps 20 inches by 2030. I will stop there.

---

## Referee Comment (RC2) · Anonymous Referee #2 · 26 Aug 2020

This article is substantially based on a previous publication of the author, Banfalvi (2017). Comparing the two, I see the same kinds of arguments and conclusions, I see very similar figures and table values. Frankly, after reading, it appears difficult to me to find what in the new paper is really new and original compared to the former one. The author should briefly mention the results adopted from other publications, and then very clearly start formulating the core of the new article with new arguments and novel conclusions. As the Abstract says, the article's topic is that "The escape of hydrogen through the planetary air leak cannot be reliably judged. Our estimation of global freshwater loss used another approach based on the sea level changes that continuously fluctuated over geological time." However, a quick

look at Wikipedia, https://en.wikipedia.org/wiki/Atmospheric_escape, shows that a current estimate of gravitational escape is 3 kg/s of hydrogen. The author neither critically reviews such previous estimates nor even mentions them; apparently, he simply ignores those by declaring them "not be reliably judged". He does not even attempt to estimate their uncertainties or discuss their weaknesses. On the other hand, his own conclusion does neither come up with an alternative new quantitative estimate for the escape rate, nor with any demonstration of reduced uncertainties by his suggested approach. The numerical result presented in the conclusion is the fluctuation of global sea level, adopted from a figure of another Wikipedia article, https://en.wikipedia.org/wiki/Past_sea_level#/media/ , in which two different estimates indicate vast uncertainties. But, in contrast to the author's claim, sea level changes cannot simply be associated with gravitational escape rates. Sea level is not immediately related to water volume as long as it is unclear to what datum the level gauge did refer to in the very past. Sea level may change at constant volume also by dynamical changes of the hypsometric curve, due to continental drift or sea-floor deformation, due to earthquakes or emerging islands such as Iceland or Hawaii. And, in turn, even volume is not proportional water mass, as thermal expansion may change their mutual relation depending on paleo-temperatures, and ice may form and be deposited on land. The water mass at the Earth's surface may also change as a result of volcanic ejections, by continental subduction or by impact of icy celestial bodies, just to mention a few aspects beyond gravitational escape. Radioactive processes deep below the surface may produce additional protons. The Earth's water inventory is very effectively protected by the "cold trap" tropopause, before surface water may escape from the stratosphere or above. Are transfer rates across the tropopause consistent with the author's loss rates? Partially compensating the loss of H+ ions, Earth is also receiving protons from the solar wind and cosmic radiation, as beautiful Auroras visibly prove. Evidently, there are numerous sources of uncertainties involved before sea level estimates may be considered as reasonable proxies for atmospheric escape rates. The author may want to carefully review and assess all such relevant processes in order

to make his claims more plausible and, ultimately, also quantitatively available. As a final remark, the quantity and unit of "osmolality" may be common in physiology and medicine, but it is not among the SI units. Papers in Ocean Science should strictly prefer SI units, however.

---

## Author Comment (AC1) · 24 Sep 2020

Point-by-point answer to Referee 2

1.  **Referee**: This article is substantially based on a pervious publication of the author (2017).
    - **Answer**: Figure 3 of the previous publication contained only two sea volumes at -50 m and +130 m sea levels. The plotting of the calibration curve was based on the measurements of Exxon's and Hallam's sea levels.
    - In the recent MS the calibration curve (Fig. 2) contains sea volumes at -50, +130 and +350 m sea leves relative to the present one, plus the highest volume converted to sea level of the infant Earth (Pope et al., Rosing et al.) that made estimations more reliable and the calibration curve could be extended to higher see levels and volumes.
    -

2.  The article topics is that the escape of hydrogen through he planetary leak cannot be reliably judged.
    - The information requested by the reviewer is found in the Introduction ( page 3).

      Lines 69-78 are quoted: " The shrinkage of freshwater reservoirs including atmospheric moisture (snow, rain, clouds) and ice (glaciers, polar ice, ice sheets, permanent snow) is contributed by the freshwater loss of global warming. The atmospheric escape of hydrogen on Earth is assumed to result in approximately 3kg/s loss of hydrogen and about 50 g/s loss of helium (Zahnle, 2006; Catling and Zahnle, 2009).The abundant $CH_4$ originating from methanogenesis could have supported the escape of hydrogen to space by orders of magnitude faster than today (Catling et al., 2001).Our own repeated calculations showed that under the conditions that exist today, the H escape could have resulted in only about 0.02% loss of the recent ocean volume.The explanation to this negligible loss could be that the escape of water: a) was much faster earlier, b)came from different sources that could sum up, or c) was not significant during the evolution of Earth.

      Regarding oxygen, only small quantities were found to escape to the Moon from the Earth (Terada et al., 2017). $O_2$ produced by photosynthesis absorbed in oceans and seabed rocks started to gas out about 1,850 Mya but was absorbed mainly by

land surfaces. When the ocean saturated, oxygen began to accumulate in the atmosphere (Holland, 2006)".

3. See level is not immediately related to water volume as long as it is unclear to what datum the level gauge did refer to in the past.
   - Data were obtained from references. The validity of data of others. were not questioned.
   -
   - **4** .The quantity and units of Osmolarity is not among the SI units in Ocean Science.
   - The osmotic concentration is the measure of solute concentration used when the solution contains more or many dissolved particles 1 Osm = $10^{23}$ dissolved particles /L solution, often not even known the all the components. The osmolarity can be calculated, by adding the molarities of the constituent ions. The composition of the seawater varies around the world and mostly only the osmolarity of the NaCl is given which is about 1000 mmol/kg water. Osmolaty is measured by an osmometer based on t he freezing point depression giving the results in osmolality Osm/kg solute or in osmolarity Osm/L. Plasma osmolality isnormally given in Osm/L. The ocen is regarded as the largest osmolyte system, thus its concentration was given in osmotic concentratio,similrly to the oamlarity of blood serum.
   -
   - **5**) Sea volume is not proportional to water mass, due to thermal expansion.
   - -The usually quoted value of the average temperature of water in the cean is by Michael Pilcher oceanographer 3.52 dedrees Celsius updated by Ken Miller MIT, Apr. 15, 1982 and supported by 175 answers and answer views. In the recent decades the deep water the temperature has slightly elevated probably in less than few hundreds of of one degree. These changes are unlikely to significantly influnce the volume of the incompressible liquid water
   - **6**. To the questions how gravitational, volcanic, radioactive processes, solar winds could have changed the sea levels and volumes of sea the author does not know the answer.

---

## Editor Comment (EC1) · Trevor McDougall (Editor) · 2 Oct 2020

Editor's Comment on the Technical Note: Estimation of global loss of freshwater based on sea level changes over geological time, by Gaspar Banfalvi

On the basis of the two reviews of this manuscript, and your reply to referee #2, I do not recommend that you prepare a re-submission, since the manuscript, if it were to be publishable in Ocean Science, would be sufficiently different to the present manuscript as to warrant it being considered as a different manuscript.

To the comments of reviewers #1 and #2, I add the following comments.

[Figure]

(1) The discussion of sea level rises of 15-25 m by 2030, 86 m by 2060, 79-155 m and possibly 206 m in the longer term (lines 217-219) are simply not credible. These estimates of sea level rise are orders of magnitude larger than the generally accepted range, as distilled by the IPCC. It is not credible for these numbers to be mentioned without at the same time saying why you think that all the other scientists around the world have got their predictions so wrong for so long.

(2) Mention is made of the changing salinity of the oceans with time, but no mention is made of the usual reason for this, namely that the salt comes from the small concentration of dissolved material arriving in the ocean from rivers.

(3) Much of the reason for the manuscript appears to be the observation that sea level may have been 500 m or 750 m higher in the past, but there appears to be no discussion of the uplift of land masses due to plate tectonics over these millions of years, such uplift may affect to proxy data that is used to infer this rise. There is a huge amount of literature on this subject of continental vertical movement which has not been cited, and an author comes to mind in this regard; K. Lambeck.

(4) For time scales of 100 million years in the past, there are now reconstructions of the shapes of continents and ocean depths. Surely the changing volume of the ocean, as given by spatially integrating the depth of the sea floor, is the first thing to consider, along with any continental uplift. The reconstruction to which I refer has been done in the lab of D. Muller, and is known as the Virtual Earth Laboratory. Why was the changing depth of the sea floor, over 100 million years or so, not mentioned? Rather, table 1 and much of the text discusses aspects of the geometry of the present ocean.

Any one of the above four points would be sufficient reason for rejection from being published in Ocean Science. As editor, I can also say that none of the reviewers that were suggested on submission to OSD agreed to review this OSD preprint. This led to a longer delay than usual, as this editor needed to find other reviewers.

---

## Author Comment (AC2) · 21 Oct 2020

Final response to Referees on Technical Note os-2020-42

Final response to Referee 1

1. Referee 1: It is difficult to make sense of this manuscript. - Answer: "My initial idea after reading the title of your manuscript was that water cannot be lost. After a closer look at your paper, I see that it offers an interesting view on processes I never thought about" (Opinion of Erwin Zehe Executive Editor of Earth System Sciences). 2. Unfortunately, I cannot even summarize the main purpose of this paper. - The main purpose of

this technical note is included in the title: "Estimation of freshwater loss based on sea-level changes over geological time. 3. Changes in sea level up to 750 meters are to be related to . . . I am not sure. - Evolutionary changes in sea levels and their converted data to sea volumes were used to calculate the loss of freshwater on Earth. In ice ages, the volume of freshwater in the form of ice, snow, glaciers increases and can be judged by the thickness of ice, snow, etc. During interglacial periods the melting of ice takes place and the increase of sea volume causes sea level rises. As the osmolarity of sea increases, less and less water is evaporated and the volume of freshwater decreases. 4. Eventually, a reference is made to the ocean being a "dynamic osmolyte" in an on-line scam journal. - I have taken over the medical chemistry lectures as an associate professor in 1990. My first lecture was on dilute solutions (vapor pressure, Raoult's law, osmolarity) and have published my first paper on osmolarity and called the ocean as a dynamic osmolyte system in Biochemical Education now known as Biochemical and Molecular Biology Education (IF 0.947). - The idea that the nearly uniform osmolarity of the blood of land vertebrates reflects the osmolarity of an ancient stage, namely the concentration of the primordial ocean at the time of migration to land is not new (Smith, 1943, 1944). As the osmolarity of the present-day ocean is 1.09 Osm, a corol-lary of my hypothesis was that the salinity of oceans is gradually increasing (Banfalvi G, 1991. Evolution of osmolyte systems. Biochemical Education. 19, 136-139). - Thes following quotations are from the book on Homeostasis – Tumor – Metastasis page 12, by Banfalvi G – 2013, Springer-Science, Dordrecht, ISBN 978-94—007-7335-6 (eBook), ISBN 978-94-007- 7334. - "When discussing osmolytes one cannot escape the thought that living cells consuming and producing substances in a dilute aqueous milieu, resemble the ultimate dilute solution, the primordial ocean" (p. 9). - " In this context, the ocean as the oldest and most important electrolyte system is discussed concerning osmotic systems of biological origin" (p. 10). - "Salinity changes of the sea over geological ages provide evidence for a dynamic osmolyte system against a persisting general geochemical balance. Fluctuations in sea salinity explain wet and dry climatic periods and are among the major driving forces of biological evolution.

The recent thermal expansion of sea volume, snowpack reduction, melting of sea ice, icebergs, and ice sheets, is a reflection of a short term dilution period, the oscillation of which is temporarily outweighing the long-term salination of the sea". - See also subsection 1.3. of this book devoted to "Global aspects of the ocean as an osmolyte system". 5. The claim that the global sea levels could rise 20 m by 2030... - Thanks to the reviewer for noticing this mistake. The reference to United Sea Level Rise Projection is given only as a home page. Unfortunately, I have taken recent sea level rises in meters but in the Projection they were given in inches. The removal of the last paragraph before Conclusions eliminates the United Sea Level Rise Projection and the removal of mixing up meters with inches solves the problem the reviewer noticed, as the recent sea-level rise is not at the focus of this study.

Final response to Referee 2 1. This article is substantially based on a pervious publication of the author (2017). - The reviewer is correct. Figure 3 of the first publication contained only two sea volumes at- 50 m and +130 m. The plotting of the calibration curve was made possible by the measurements of of the Exxon's and Hallam's sea levels. In the new MS the calibration curve (Fig. 2) contains sea volumes at -50, +130 and 350 m sea leves relative to the present one plus the highest sea level of the infant Earth (Pope et al., Rosing et al.) that made the judgments more reliably and the calibration curve could be extended to higher see levels and volumes. 2. The article topics is that the escape of hydrogen through he planetary leak cannot be reliably judged. - The loss of hydrogen, helium, methane and oxygen is dealt with in the Introduction page 3,containing our own repeated calculations showing that under the conditions that exist today, the H escape could have resulted in only about 0.022009, Catling 2001; (Terada et al., 2017; Holland, 2006. The information requested by the reviewer in found int he Introduction ( page 3) E.g. lines 71-78: " 3.See level is not immediately related to water volume as long as it is unclear to what datum the level gauge did refer to in the past. - Data were obtained from references. Their validity. were not questioned. - 4.The quantity and units of Osmolarity ( Osm, mOsm) in Ocea Science is not among SI units. The osmotic concentration is the measure of solute concentration used when

the solution contains more or many dissolved particles 1 Osm = 1023 dissolved particles /L solution, often not even known the all the components. The osmolarity can be calculated, by adding the molarities of the constituent ions but the composition of the seawater variesaround the world and mostly only the osmolarity of the NaCl is given which is about 1000 mmol/kg water. Osmolaty is measured by an osmometer based on t he freezing point depression giving the results in osmolality Osm/kg solute or in osmolarity Osm/L. Plasma osmolality isnormally given in Osm/L. The ocen is regarded as the largest osmolyte system thus its concentration were given in osmotic concentrations. - 5) See volume is not proportional to water mass, due to thermal expansion. These changes may modulate sea levels and volumes but unlikely to significantly influrnce them, whereas their biological impact may be significantespecially in transition periodes. - 6. To the questions how gravitational, volcanic, radioactive processes, solar winds could have changed the sea levels and volumes the author does not know the answer.

Final response to Editor 1.Editor: The author of OS-2020-42 says that he is unable to add a comment on the remarks of review 1, and he would also like to respond to my editorial comment. Would you please open up the time window for this manuscript, to enable him to do this? (Editorial Comment Oct. 8, 2020)". - The answer of the author to the editor: To the contrary what the editor states the author was able to comment on the remarks of reviewer 1 on June 30, 2020, and could attach the comment to the interactive comments (https://doi.org/10.5194/os-2020-42-RC1, 2020) The author could not find this comment in the discussion panel but insisted and asked several times the editorial office of OS to attach it because the electronic accessibility was paralysed. The author gave the authorisation for the attachment of this comment two times - The comments to reviewer 1 and 2 were repeatedly submitted to the editorial office of OS. Finally, the comment to reviewer 1 was uploaded by the editorial office in the form of a supplement: https://os.copernicus.org/preprints/os-2020-42/os-2020-42-AC1-supplement.pdf. -Similarly, my comment to referee 2 could not be placed into the interactive discussion and was added by the editorial office in the

form of a supplement as https://os.copernicus.org/preprints/os-2020-42/os-2020-42-AC-supplement.pdf. These hidden supplements were probably downloaded not many times by the readers of OS contradicting the rule of accessibility of data. The lack of response by the editorial office between Sept. 5 and Oc.2 as the end of interactive discussion (Sept. 26) was approaching forced the author to contact first the editorial board then the executive editors (including the editor). 2. After the interactive discussion period was over the chief executive editor asked the author on behalf of all the executive editors to submit all responses, revised manuscripts etc. through the Copernicus Editorial system as with the original document (Sept. 30). - The revised MS was not submitted to OC as the responding editor arbitrarily decided to extend the interactive discussion period and send his comment (Oct. 2). 3. In this comment, quoting the editor:" The author of OS-2020-42 says that he would also like to respond to my editorial comment". - Besides that, point-by-point answers were given to reviewers 1, 2 and the editor, no responses were obtained whether my answers were accepted or denied by the reviewers. The lack of responses in my reading means that my answers were accepted. Serious mistakes were made In the reviews by the referees. These factual errors were not acknowledged (see files containing answers to the reviewers). 4. The editor admitted that his interactive discussion was a rejection, but offered his willingness to receive further arguments, - The editor made his arbitrary decision to reject the MS before the discussion period was over. Answers to further questions: 5. "The MS if it were publishable in Ocean Science, would be sufficiently different from the present manuscript as to warrant it considered as a different manuscript". - In answer to reviewer 1 and in the letter sent to executive editors author suggested that only one paragraph would be removed that was criticised by reviewer 1. An external reviewer suggested the removal of the paragraph related to the recent sea-level rise, but not relevant here. Consequently, the resubmitted MS would not be a new one. 6. The discussion of the sea-level increase is not credible. - The author has acknowledged this mistake to reviewer 1 and suggested to omit this paragraph. The MS deals with the highest sea-levels and sea volumes belonging to these sea-levels. Water is likely to

have been present in other small inner planets (Venus, Mars) but disappeared over geological ages. 7. "Mention is made of the changing salinity of the oceans with time, but no mention is made of the unusual reason for this". - Thanks to the reviewer as salinity indeed deserves mention. The ocean was described as a dynamic osmolyte system. Three publications are given in the recent MS to OS, including Banfalvi G. Biochemical Education 19, 136-139, 199. The salinity of the ocean could have been contributed among others by a) continental drift and outpouring of lava increasing the surface of the seabed, b) weathering and denudation carrying away the surface of the land, deposited in the ocean building sedimentary rocks, c) One pathway of the hydrologic cycle is the flow of freshwater of that carries diluted salt to the sea. d) Loss of water to space, e) Chemical pollution contributed by man. The following quotation is taken from my answers given to reviewer 1, in the interactive discussion:" The nearly uniform osmolarity of the blood of land vertebrates reflects the osmolarity of an ancient stage namely the concentration of the primordial ocean. . .." For the sake of better understanding in the final version of MS, more details of the global osmolyte system will be given. 8." Eventually, a reference is made to the ocean being a "dynamic osmolyte" in an online scam journal.Lectures in the Medical University, Budapest on dilute solutions (vapour pressure, Raoult's law, osmolarity) resulted in my first paper on osmolarity in Biochemical Education (now Biochemical and Molecular Biology Education. This Journal was established by Edward Wood at the University of Leeds, and is not and was not a scam journal). In this journal, the ocean was mentioned as the most extensive dynamic osmolyte system on Earth. The idea that the nearly uniform osmolarity of the blood of land vertebrates reflects the osmolarity of an ancient stage, namely the concentration of the primordial ocean at the time of vertebrates migration to land is not new (Smith, 1943, 1944). As the osmolarity of the present-day ocean is 1.09 Osm, a corollary of my hypothesis was that the salinity of oceans is gradually increasing (Banfalvi G, 1991. Evolution of osmolyte systems. Biochemical Education. 19, 136-139). - The following quotations are from the book Banfalvi G: Homeostasis – Tumor – Metastasis page 12, 2013, Springer-Science, Dordrecht, ISBN 978-94—007-7335-6 (eBook), ISBN 978-

94-007- 7334: " When discussing osmolytes one cannot escape the thought that living cells consuming and producing substances in a dilute aqueous milieu, resemble the ultimate dilute solution, the primordial ocean" (ibidem p. 9). " In this context, the ocean as the oldest and most important electrolyte system is discussed concerning osmotic systems of biological origin" (ibidem p. 10)." Salinity changes of the sea over geological ages provide evidence for a dynamic osmolyte system against a persisting general geochemical balance. Fluctuations in sea salinity explain wet and dry climatic periods and are among the major driving forces of biological evolution. The recent thermal expansion of sea volume, snowpack reduction, melting of sea ice, icebergs, and ice sheets, is a reflection of a short term dilution period, the oscillation of which is temporarily outweighing the long-term salination of the sea". - See also subsection 1.3. of this book devoted to" Global aspects of the ocean as an osmolyte system". 9." I have advised against preparing a revised manuscript because any manuscript that would be acceptable for publication in Ocean Science would need to be the result of much extra research on topics that had not been addressed in the present manuscript. In this way, any acceptable manuscript would be essentially a new one. - The MS is only a note. OS did not ask to verify water loss (which would be impossible). The author restricted himself to deal with volumetric changes in the ocean. The data obtained by the calculations presented in the MS correspond to the results of others. By means of evolutionary high sea-levels and volumes a calibration curve was plotted. Our calculations confirmed that the volume of sea (1332.9 x 106km3) is practically the same as the autoritative value (1332.4 x 106 km3) of Charette and Smith (2010). The following new data were provided by the MS : - sea-level and volume of the Infant Earth, - the volume of globally lost water, - volume of the continental shelf, - volume of underwater features without the continental shelf. Our calculations showed that the volume of the underwater features excluding the continental shelf represents only 0.45 There are no other data available at the moment that would reflect the volume of freshwater loss on Earth. Our calculations are verifiable, and supported by 30 references- The author is confident that the new approach and new data are sufficient for the Note to be published in OS. 10. Question of the Editor:" Why do you think that all the other scientists around the world have got their predictions so wrong for so long". - The author is not the only one who thinks that the Earth is losing water. (See in the Introduction" page 1, lines 60-62, Schröder et al. 2008). From the 3kg/sec loss of hydrogen (Zahnle, 2006; Catling and Zahnle, 2009; Catling et al. 2001, lines 68-75) we have calculated with Prof. Henry Paulus (Harvard Medical School) repeatedly that the hydrogen escape could have caused only about 0.02 11.After reconstructions of the shapes of continents and ocean depths change. The depth of the seafloor impacts sea volume. - This would be true if not the average sea dept would have been used for calculations. Fig. 1a demonstrates how sea volumes were calculated from the average of radii of geoid Earth and the average depth of the sea.

12. Editor:" The author would also like to respond to my editorial comment" (Oct. 8, 2020). - Summary was given to this editorial comment on Oct. 9. The editor admitted that his interactive discussion was a rejection but offered his willingness to receive further arguments. Still, his refusal means that these arguments would not matter. - The author expressed his willingness to answer further questions and repeatedly asked to reflect on the answers he gave to the reviews. As the editorial office and referees gave no answers, there was no interactive discussion, Thus the original answers are the final answers to referees 1 and 2, The author answered all the question to the best of his knowledge. The ideas related to the global loss of water and the global osmolyte system have been published and his priority regarding publication secured. The importance of this Note is that it shows how to calculate global water loss and is the "icing on the cake". Ocean Science will be proud to have published.

Please also note the supplement to this comment:
https://os.copernicus.org/preprints/os-2020-42/os-2020-42-AC2-supplement.pdf

---

## Editor Comment (EC2) · Trevor McDougall (Editor) · 21 Dec 2020

This OSD manuscript will not be published in Ocean Science.

The manuscript and its revision falls below the acceptable standard of methodology for a scientific journal. The manuscript assumes a spherical earth with a constant area fraction of land/ocean and a flat ocean floor. However, earth science has advanced remarkably over the past 50 years, and we now have some idea of how the continents have evolved over hundreds of millions of years, how these contents have moved around the planet, we now know that the oceans have mid-ocean ridges, and we know that there is eustatic rebound and uplift of relative sea level at measuring sites

(e.g. corals). All of these things impact on how observations of relative sea level at a series of sites affect the reconstructed estimation of global ocean volume. A scientific paper that discusses the change in ocean volume over hundreds of millions of years would need to take this knowledge into account in its numerical computations. Hence this OSD preprint is not accepted for publication in Ocean Science.